# A Glycoprotein-Based Surface-Enhanced Raman Spectroscopy–Lateral Flow Assay Method for Abrin and Ricin Detection

**DOI:** 10.3390/toxins16070312

**Published:** 2024-07-11

**Authors:** Lan Xiao, Li Luo, Jia Liu, Luyao Liu, Han Han, Rui Xiao, Lei Guo, Jianwei Xie, Li Tang

**Affiliations:** 1Key Laboratory of Ethnomedicine (Minzu University of China), Ministry of Education, School of Pharmacy, Minzu University of China, Beijing 100081, China; xlzcfhl16@163.com (L.X.);; 2Laboratory of Toxicant Analysis, Academy of Military Medical Sciences, and State Key Laboratory of Toxicology and Medical Countermeasures, Beijing 100850, China; 3Guangdong Lifotronic Biomedical Technology Co., Ltd., Dongguan 523808, China; 4College of Pharmacy, Hebei Science and Technology University, Shijiazhuang 050018, China; 5State Key Laboratory of Pathogen and Biosecurity, Academy of Military Medical Sciences, Beijing 100071, China

**Keywords:** surface-enhanced Raman scattering, lateral flow assay, nanotag, abrin, ricin, glycoprotein

## Abstract

Abrin and ricin, both type II ribosome-inactivating proteins, are toxins of significant concern and are under international restriction by the Chemical Weapons Convention and the Biological and Toxin Weapons Convention. The development of a rapid and sensitive detection method for these toxins is of the utmost importance for the first emergency response. Emerging rapid detection techniques, such as surface-enhanced Raman spectroscopy (SERS) and lateral flow assay (LFA), have garnered attention due to their high sensitivity, good selectivity, ease of operation, low cost, and disposability. In this work, we generated stable and high-affinity nanotags, via an efficient freezing method, to serve as the capture module for SERS-LFA. We then constructed a sandwich-style lateral flow test strip using a pair of glycoproteins, asialofetuin and concanavalin A, as the core affinity recognition molecules, capable of trace measurement for both abrin and ricin. The limit of detection for abrin and ricin was 0.1 and 0.3 ng/mL, respectively. This method was applied to analyze eight spiked white powder samples, one juice sample, and three actual botanic samples, aligning well with cytotoxicity assay outcomes. It demonstrated good inter-batch and intra-batch reproducibility among the test strips, and the detection could be completed within 15 min, indicating the suitability of this SERS-LFA method for the on-site rapid detection of abrin and ricin toxins.

## 1. Introduction

Abrin and ricin are protein toxins extracted and isolated from the seeds of *Abrus precatorius* and *Ricinus communis*, respectively. These two plant-derived toxins have similar structures and functions. They are classified as type II ribosome-inactivating proteins (RIP IIs), composed of two polypeptide chains, A and B, linked by a single disulfide bond. The A chain functions as the effector, and the B chain acts as the binding component. Several other RIP-II proteins from plants exhibit similar structures and cytotoxic mechanisms to those of abrin and ricin. For example, pulchellin, derived from *Abrus pulchellus* [1], a close taxonomic relative of *Abrus precatorius*, also has a highly similar structure to abrin. Additionally, abrus agglutinin (AAG) is also found in the seeds of *Abrus precatorius* alongside abrin, together constituting about 0.15% of the total seed weight [2]. Furthermore, ricin and Ricinus communis agglutinin 120 (RCA120) coexist in the castor bean, making up 1–5% of its total weight. AAG and RCA120 are heterotetrameric proteins with a molecular weight of about 120 kDa. However, both agglutinins present a lower toxicity than the concurrent toxins. To illustrate these structural homologies, the three-dimensional structures of abrin, ricin, pulchellin, AAG, and RCA120 are shown in Figure 1.

Both abrin and ricin are regulated by the International Chemical Weapons Convention and the Biological and Toxin Weapons Convention due to their potential to threaten human health and societal security. Given their high toxicity, as the median lethal dose(LD_50_) values of abrin and ricin (i.v., mice) are 0.7 μg/kg and 3 µg/kg [4], their ease of production, the accessibility of raw materials, and the lack of effective antidotes, there is an urgent demand for the development of efficient, sensitive, and rapid detection methods, with a preference for those that can be employed in the field.

Among all rapid on-site detection techniques, gold nanoparticle (AuNP)-based immunochromatographic lateral flow assays (LFAs) represent one of the prominent candidates. With the antibodies prevalently employed as affinity modules, the LFA has been widely applied in toxin detection for its simplicity and speed. However, it is occasionally limited by its moderate sensitivity. For example, the limit of detection (LOD) of LFA for abrin in buffer solutions is typically 3 ng/mL [5,6]. Some efforts to enhance sensitivity have been explored, such as silver staining [7] or upconversion fluorescence [8]; both LODs in a simple buffer were 0.1 ng/mL. It should be noted that the additional costs of LFAs include silver nitrate treatment, prolonged reaction time, increased manufacturing expenses, and temperature-liable sensitivity, which may limit their performance and stability in diverse applications.

Beyond the conventional detection modules used in LFAs, such as colorimetric and fluorometric probes or tags, there is considerable potential in surface-enhanced Raman scattering (SERS) detection modules that are adaptable to LFAs. By inducing local electromagnetic field amplification on the surface of nanostructures, the Raman response can be enhanced by several orders of magnitude in SERS [9]. Innovative nanomaterials have shown promising sensitivity, including nanocomposites [10,11], functionalized gold nanoparticles, and graphene [12,13]. Therefore, SERS-LFA attracted increased attention when it was created in 2007 [10]. SERS-LFA has been successfully applied to detect a variety of analytes, including, but not limited to, proteins, bacteria, viruses, nucleic acids, antibiotics, and other biomolecules and microorganisms [11]. For example, in 2016, Hwang et al. [12] reported a pioneering SERS-LFA biosensor for the highly sensitive and rapid detection of Staphylococcal Enterotoxin B in buffered samples, achieving an LOD of 0.001 ng/mL. In 2022, Jia et al. [13] tested the feasibility of the SERS-LFA method for ricin detection through a sandwich immunorecognition strategy, achieving an LOD as low as 0.1 ng/mL.

The most critical and flexible aspect of SERS-LFA is the design and selection of its recognition elements. These elements include immuno-SERS tags [14,15], nucleic acid-based SERS tags, including aptamers [16], clustered regularly interspaced short palindromic repeat (CRISPR) tools [17], and glycoproteins [18]. Among these, antibodies are the most commonly used, despite drawbacks such as considerable batch-to-batch variability and high costs. Glycoproteins offer an alternative as affinity recognition elements characterized by their high affinity, good stability, and cost-effectiveness. Herein, we propose an SERS-LFA method employing glycoproteins for affinity recognition. Using the galactose-binding properties of the B chain of abrin and ricin, we fabricated a sandwich format using asialofetuin (ASF) and concanavalin A (ConA) in the SERS-LFA test strip. These test strips enable the rapid and highly sensitive detection of both abrin and ricin. They have been effectively applied to the screening of white powder samples, juice drinks, and plant specimens. We anticipate that this SERS-LFA assay will provide potential support in developing countermeasures against biological threats.

## 2. Results

### 2.1. Schematic Diagram of the SERS-LFA

Figure 2 schematically represents the SERS-LFA. Functionalized with the Raman reporter molecule 5,5′-dithiobis-(2-nitrobenzoic acid) (DTNB), AuNPs are conjugated with the glycoprotein ASF through a PEG linker, creating an affinity element that we named “SERS nanoGPtags”, representing SERS nano glycoprotein tags. These tags are loaded onto the conjugated pad and assembled with other components to complete the assembly of the test strip. During detection, the test strip is immersed in the diluted sample solution and incubated for 15 min. If a toxin, either abrin or ricin, is present in the sample solution, then it will interact with the ASF on the SERS nanoGPtags, subsequently captured by the glycoprotein ConA in the T line and the ASF antibody in the C line, resulting in the formation of two distinct bands. Without toxins, the SERS nanoGPtags are only captured by the C line. After the incubation, the test strip is removed from the solution and air-dried at 37 °C, and the ERS signal from the T line is collected using a portable SERS spectrometer.

### 2.2. Preparation and Characterization of SERS NanoGPtags

Preparing robust and facile SERS tags is pivotal for constructing a sensitive SERS-LFA method. Initial attempts using a standard 1-(3-Dimethylaminopropyl)-3-ethylcarbodiimide (EDC)-N-Hydroxy succinimide (NHS) coupling method to fabricate SERS tags with AuNPs proved ineffective. Inspired by DNA functionalization techniques for AuNPs [19], we adopted a covalent coupling strategy via a PEG linker [20] and successfully implemented a freezing method. This approach facilitated the covalent attachment of the glycoprotein ASF, which has a flexible structure, onto the surface of AuNPs. To minimize steric hindrance, the small-molecule DTNB was conjugated onto the surface of AuNPs initially, then succeeded by ASF to form AuNPs@DTNB@ASF, i.e., SERS nanoGPtags (scheme shown in Figure 3A).

For AuNPs@DTNB, the solution changed from purple-red to colorless after a two-hour freeze, indicating that AuNPs@DTNB easily agglomerated under frozen conditions. In contrast, the SERS nanoGPtag maintained its purple-red color throughout the freeze–thaw cycle, and the SERS signal of DTNB was enhanced after freezing (Figure 3B). This enhancement suggests that glycoproteins provide good protection and enhanced stability against agglomeration induced by −20 °C. A possible reason is that, in the freezing period, the flexible structure of ASF glycoprotein will adapt its shape by stretching and rearranging in order, as ice crystals form and melt, leading to a more robust orientation on the surface of AuNPs. This rearrangement enables DTNB to function effectively as a Raman reporter within the gaps of the covalently attached ASF, thereby ensuring the stability and sensitivity of the SERS nanoGPtags.

We characterized AuNPs and SERS nanoGPtags using ultraviolet–visible spectroscopy (UV–Vis), Zeta potential measurements, and transmission electron microscopy (TEM). The plasma absorption wavelengths of the prepared original AuNP solution, the functionalized AuNPs@DTNB, and the SERS nanoGPtags (i.e., AuNPs@DTNB@ASF) were 531 nm, 533 nm, and 537 nm, respectively. The slight red shift indicates the successful coupling of DTNB and ASF to the surface of AuNPs (Figure 4A).

TEM characterization (Figure 4B,C) revealed that AuNPs@DTNB are spherical, well dispersed, and relatively monodisperse, with an average particle diameter of 42.8 ± 3.4 nm. The SERS nanoGPtags (Figure 4D,E) are elliptical, exhibit slightly reduced dispersion, and have an average diameter of 41.2 ± 4.7 nm.

The absolute value of the Zeta potential represents the stability of the system, and the positive and negative value signifies the charge on the surface of the nanomaterials. The Zeta potential measurement results (Figure 4F) show that the AuNP surface carries a negative charge, with an initial value of −14.3 mV. Upon conjugation with DTNB and ASF, the Zeta potential further shifts to more negative values, −37.2 mV and −39.3 mV, respectively, suggesting enhanced stability.

### 2.3. Optimization of Experimental Conditions for SERS-LFA Strips Prepared by the Wet Method

In our preliminary assessment of the SERS-LFA method, we determined that the optimized concentration for the T line drawing membrane was 0.5 g/L (Appendix A). Concentrations exceeding 0.7 g/L for the blank control T line led to false-positive test bands, while lower concentrations reduced the sensitivity.

The optimized amount of SERS nanoGPtags for the wet test strip was 2 μL (Appendix A). Exceeding 2 μL caused false-positive bands in the blank control T line. Conversely, with only 1 μL of SERS nanoGPtags, the T line could not visibly detect the corresponding band even at a concentration of 100 ng/mL abrin. Although an SERS response can be measured, the intensity was relatively low. Hence, we confirmed that the optimized volume of SERS nanoGPtags was 2 μL.

By assessing the SERS response intensity at 1331 cm^−1^ as indicative of the laser power and integration time, we determined an optimized collection time of 15 s with a laser power setting of 150 mW (Appendix A). When the laser power reached 180 mW, the strip would be ablated.

### 2.4. Sensitivity of the Two Toxins with SERS-LFA Strips Prepared by the Wet Method

Figure 5A displays the visual results for SERS-LFA test strips for detecting abrin and ricin. At concentrations below 30 ng/mL, the T line was faint to the naked eye due to a small aggregation of SERS nanoGPtags. Figure 5B presents the SERS signals at the T line of different concentrations of toxins, addressing the low sensitivity of visual detection. Calibration curves were constructed using the intensity of the main peak at 1331 cm^−1^ versus the concentration of toxin, yielding a correlation coefficient of 0.971 for abrin and 0.939 for ricin (Figure 5C). The LODs for abrin and ricin were determined to be 0.03 and 0.1 ng/mL, respectively, corresponding to three times the intensity of the background noise at the T line position.

### 2.5. Repeatability of the Two Toxins with SERS-LFA Strips

Abrin and ricin at 100 ng/mL were examined with 10 replicate SERS-LFA test strips (Figure 6). All 10 test strips for both abrin and ricin displayed distinct bright light-red bands at the T lines, demonstrating good intra-batch reproducibility, with relative standard deviations (RSDs) of less than 5.5% for abrin and 6.5% for ricin. To assess inter-batch reproducibility, abrin at 1 ng/mL and 100 ng/mL were tested using three strips from three separate batches. The RSDs for these tests were below 6.6% and 13.9%, respectively, confirming good repeatability.

### 2.6. Optimization of Experimental Parameters for SERS-LFA Strips Prepared by Dry Method and Method Sensitivity of the Two Toxins

The wet test strip method is user-friendly, rapid, highly sensitive, and reproducible. However, it is less convenient for field applications, as most commercial test strips exist in the form of dry test strips. We thus optimized the conditions for dry test strips, focusing on the spraying volume of SERS nanoGPtags on the conjugate pads. The best volume for the SERS nanoGPtags on the pad was found to be 2.5 μL (Appendix A), which prevented false-positive bands and provided a robust SERS response. Volumes exceeding 3 μL resulted in false-positive bands in the blank control T line, while volumes below 2 μL led to reduced sensitivity.

Figure 7A,B present the visual and SERS results of the SERS-LFA test strip via the dry method for detecting the two toxins. Figure 7C shows the calibration curves fitted based on the intensity of the main peak at 1331 cm^−1^ versus the toxin concentration, with a correlation coefficient of 0.990 for abrin and 0.994 for ricin. The LODs for abrin and ricin were 0.1 and 0.3 ng/mL, respectively, indicating a threefold decrease in sensitivity compared to the wet method, yet still maintaining high sensitivity.

### 2.7. Specificity of Abrin with Specific SERS-LFA Strips

To assess specificity, we examined two RIP II agglutinins (AAG, RCA120), two sub-chains of ricin (ricin A chain, RTA; ricin B chain, RTB), one RIP I protein (saporin), two RIP-II proteins (Shiga toxins Stx1 and Stx2), and one non-RIP biotoxin (staphylococcal enterotoxin B, SEB) at a concentration 100 times the LOD (30 ng/mL). Only abrin, ricin, RTB, AAG, and RCA120 exhibited a red band at the T line (Figure 8A). Figure 8B,C display the SERS spectra measured at the T line and the histogram of the SERS signal intensity at 1331 cm^−1^, respectively. This indicates that this SERS-LFA method is specific to galactose-binding RIP II proteins.

In addition, we evaluated the specificity of the assay toward inactivated and natural abrin and ricin toxins (Appendix A). We examined two decontamination methods for abrin and ricin at a concentration of 1 μg/mL, namely: (1) treatment with 5% NaOH overnight and (2) autoclaving for 20 min at 121 °C, as per reference [21]. We found that the SERS response from natural toxins was about 1.5 times higher than that from inactivated toxins at the same concentration. Considering that the SERS intensity vs. concentration relationship exhibits a sigmoidal curve, we calculated the remaining concentration after inactivation, which was 0.13 ng/mL (abrin) and 0.13 ng/mL (ricin) in 5% NaOH overnight and 0.26 ng/mL (abrin) and 0.12 ng/mL (ricin) at 121 °C for 20 min; this indicates that our LFA assay can differentiate between active and inactive toxins to some extent. Since there was a relatively large deviation in the lower concentration in the logistic model of the fitting curve, a more accurate distinction is not yet possible. We recommend addressing this issue by combining the N-glycosidase activity of the A-chain to achieve the distinction between active and inactive toxins [22,23].

### 2.8. Examination of White Powder Samples

Abrin is a white crystalline powder that can be mistaken for other common white powders and mixed with different materials. Unknown white powders are frequently encountered in environmental sampling [24,25] and present a significant challenge in the first emergency response for suspected biological threats. We tested a set of eight white powders, as listed in Table 1, in the presence and absence of 100 ng/mL of abrin. The white powders were tested at a concentration of 0.1 g/L. The goal was to assess whether any powders would yield false-positive results for abrin in the absence of the toxin or if they would interfere with or inhibit the detection of abrin when mixed with it.

In the absence of abrin, all white powders tested negative in the SERS-LFA assay (Figure 9). In the presence of abrin, all powders except for the protein powder correctly identified abrin in the samples. Abrin was detected when the concentration of abrin in the protein powder was increased to 5 μg/mL, suggesting that substances present in the protein powder, such as ASF, may interfere with, or inhibit, the detection of abrin.

### 2.9. Measurements for Actual Botanic Samples and One Spiked Juice Sample

We chose crude abrin from jequirity beans, extracts of the peel from unripe castor bean fruits, and extracts from castor flowers as authentic botanical samples for our assays (Figure 10A). We also conducted corresponding cytotoxicity assessments (Appendix A).

For the crude abrin samples from jequirity beans, we determined the presence of abrin via the SERS-LFA test strips as 1.29 mg/g. The half-maximal inhibitory concentration (IC_50_) for the crude jequirity bean toxin was measured to be 2.98 ± 0.06 ng/mL, which was approximately 1/6 the cytotoxicity of abrin-a (IC_50_: 0.47 ± 0.19 ng/mL), as shown in Appendix A.

The ricin content was calculated to be 31.6 μg/g for the extract from the peel of immature castor beans. The IC_50_ for the peel extract of immature castor beans was determined to be 121.2 ± 31.6 ng/mL, about 1/107 the cytotoxicity of ricin (IC_50_: 1.34 ± 0.14 ng/mL), as indicated in Appendix A. The castor bean flower extracts maintained an HeLa cell viability rate of 90% at a concentration of 3 μg/mL.

We further examined an abrin-spiked peach juice drink sample. Abrin was diluted directly in peach juice to final concentrations of 1 ng/mL and 100 ng/mL for this SERS-LFA test (Figure 10B). The SERS responses of T lines were measured, with spike recovery rates of 80.04% for 1 ng/mL of abrin and 94.83% for 100 ng/mL of abrin in the peach juice drink.

## 3. Discussion

Currently, LFAs face three major challenges: inadequate sensitivity (failure to detect), insufficient accuracy, and limited detectable targets. The key factor in detection lies in the selection of the recognition element. In SERS-LFA, antibodies or aptamers are commonly used as recognition elements. However, only a few specific antibodies are available for the detection of ricin, and even fewer are suitable for abrin. Additionally, antibodies are prone to batch-to-batch variability, are expensive, and do not directly reveal information about the structure of toxins. Compared to antibodies, glycoproteins offer superior thermal and chemical stability, a simple preparation process, and cost-effectiveness. Hence, it is feasible to develop an SERS-LFA detection method based on a glycoprotein sandwich complex. Our preliminary research identified a combination of ASF and ConA glycoproteins that exhibit a high affinity for abrin and ricin. This combination can potentially enhance the detection sensitivity, thereby improving the reliability and accuracy of the assay.

The issue of inaccurate detection in LFAs often pertains to problems with quantification. Overcoming the challenge of material stability and preparing SERS nanoGPtags with consistent performance can address the issue of inaccurate quantification to some degree.

We initially considered using gold magnetic particles as the substrate, but the commercial gold magnetic particles available were unsuitable for coupling with DTNB (Appendix A). Consequently, we chose to use the more robust AuNPs. After coupling small-molecule DTNB with AuNPs, the conventional EDC-NHS activation method for ASF coupling failed, so we turned to a freezing method based on the work by Liu et al. [19]. This method is recognized as enhancing the density and regularity of ssDNA on the surface of AuNPs. Since glycoprotein also possesses a relatively robust core and flexible glycan structure, the protein shape can be readily adapted to an ordered stretching and rearranging on the surface of AuNPs during the formation and melting of ice crystals. Therefore, we adopted the freezing method to couple AuNPs@DTNB with ASF successfully.

Before freezing, we first used a PEG linker to bind ASF, creating the complex ASF-PEG7-S-S-PEG7-ASF. We then conjugated this complex to the available surface areas of AuNPs that had been partially coupled with the small-molecule DTNB via the Au-S interaction. This approach effectively addresses the challenges of steric hindrance that can occur during the conjugation of large molecules. Moreover, this approach simplifies the process by eliminating the need to optimize the reaction time and temperature, which are typical variables in traditional bioconjugation strategies.

To overcome the challenge of detecting a limited number of targets, implementing multiple T lines can facilitate the simultaneous detection of multiple toxins [26]. Our SERS-LFA method, based on a sandwich glycoprotein system, has the potential to be adapted into a universal method for detecting various plant-derived RIP II toxins that bind to galactose, such as ricin and abrin and their agglutinins, which helps to expand the range of target detection.

As detailed in Table 2, we have conducted a comparative attribute analysis of our SERS-LFA with other recently reported SERS-based methods for detecting abrin and ricin. Our method demonstrates advantageous performance over other methods in several key aspects, including stability, the spectrum of detectable toxins, and sensitivity at the ng/mL level. Moreover, our method has a comparable sensitivity to the latest microfluidic chip-based method for detecting ricin and abrin [27].

Regarding specificity, we have chosen eight different proteins. The first group consists of RIP-II proteins with similar structures, i.e., AAG and RCA120. The second group comprises RIP-II proteins with somewhat different structures, such as Stx 1 and Stx 2. The third group consists of sub-chains of ricin: specifically, RTB and RTA. The fourth group features an RIP-I protein, saporin; on the other hand, RTA is a type of RIP I protein. The last group represents another bioterrorism-related toxin, SEB.

Our results reveal the specificity of this SERS-LFA method for RIP II proteins with galactose-binding affinity, either in their entirety (abrin, ricin, AAG, RCA120) or in part, via the β-Gal binding B chain (RTB). These four plant-derived RIP-II proteins are structurally highly similar, thus yielding positive signals. Other proteins, such as RTA, Stx 1, Stx 2, saporin, and SEB, did not cross-react with this assay. We attribute the specificity of this glycoprotein sandwiched assay to glycan specificity, since both Stx 1 and Stx 2 are RIP IIs but exhibit specificity for globotriaosylceramide (Gb3, Galα1-4Galβ1-4Glc) rather than β-Gal [37]. However, for pulchellin, a relative of abrin, or viscumin, another structurally similar RIP-II protein with galactose binding ability, we have not yet obtained those substances and can only infer that those might show a positive response. Indeed, this method may produce a positive reaction for RIP-II toxins with galactose-binding specificity. Subsequently, it can be used to measure and screen more adjacent RIP-II toxins, forming a measurement matrix, and thereby making this method more universally applicable.

Since abrin and ricin are white powders that can be easily mistaken for other common white materials, they pose a considerable challenge in the first emergency response to biothreats. To address this, we spiked 100 ng/mL of abrin into the solutions of eight different white powders, and our method successfully detected abrin in all but the protein powder solution. When the concentration of abrin was increased to 5 μg/mL, the detection was achieved even in the protein powder solution, demonstrating the applicability of our method for the rapid and effective detection of toxins in complex matrices.

Furthermore, we tested authentic samples such as crude abrin, extracts from the peel of unripe castor beans, and extracts from castor flowers. The higher the amounts of the toxin, the higher the cytotoxicity, indicating that our SERS-LFA assay is suitable for measuring active toxins. Specifically, the ricin content in the castor flowers was low or absent, suggesting that ricin was primarily located in the seeds of the castor plants.

We also achieved positive results with satisfied recovery at two concentrations for peach juice drinks spiked with abrin.

The aim of this method is not to differentiate abrin from ricin, or vice versa. If a need arises to distinguish abrin and ricin further in the future, corresponding magnetic beads can be used to capture abrin and ricin before detection. Alternatively, we could incorporate additional T lines composed of abrin or ricin antibodies into our SERS-LFA method, facilitating their differentiation.

## 4. Conclusions

In this research, we have generated novel SERS nanoGPtags with high affinity and stability through a simple yet effective freezing method for the first time. These tags have been successfully employed to construct a new SERS-LFA test strip based on a glycoprotein sandwich system, enabling the rapid and sensitive detection of abrin and ricin. This approach offers a promising alternative to immuno-strips through its enhanced stability and improved repeatability.

Building upon the good sensitivity and reproducibility of SERS-LFA strips, we achieved the LODs of 0.03 and 0.1 ng/mL for abrin and ricin via the wet method, and 0.1 and 0.3 ng/mL for abrin and ricin via the dry method. Considering that the LD_50_ for ricin in humans after administration by inhalation or injection is 7–10 μg/kg [38], i.e., 7–10 ng/mL, our dry method LOD of 0.3 ng/mL allows for the detection of a 0.03–0.04 LD_50_. Additionally, the detection process can be completed within 15 min, making it suitable for rapid on-site detection.

This method has also been successfully applied to detect eight types of white powder, one fruit juice, and three plant samples. Except for protein powder, most white powders were well recognized with or without abrin, indicating that this method suits toxin detection in complex matrices. We also measured three botanical samples and one spiked juice sample and achieved satisfactory results.

Considering the specificity and applicability of the tested samples, we suggest that this detection method can be implemented on a large scale for the primary screening of the test strip and detection equipment. It could be used to rapidly screen unknown white powders and plant-toxin-derived substances and to quickly exclude bacterial toxins such as SEB and botulinum toxin. We hope that this convenient glycoprotein-based SERS-LFA assay can meet the first emergency response demands in terms of biosecurity.

## 5. Materials and Methods

### 5.1. Chemicals and Materials

Ricin, abrin, RCA120, AAG, Stx1, Stx2, RTB, and RTA were all milligram-grade and produced in the laboratory under the regulations of biosafety grade 2 guidelines. SEB was kindly donated by Prof. Yongqiang Jiang in the State Key Laboratory of Pathogen and Biosecurity, Academy of Military Medical Sciences, China. Polyclonal antibodies against ASF [NB100-62357] were purchased from Novus Biologicals USA (Centennial, CO, USA). Chloroauric acid, PEG NHS ester disulfide (4,7,10,13,16,19,22,25,32,35,38,41,44,47,50,53-hexadecaoxa-28,29-dithiahexapentacontanedioic acid di-N-succinimidyl ester, NHS-PEG-S-S-PEG-NHS, n = 7, MW 1109.26), DTNB, 4-(2-hydroxyethyl) piperazine-1-ethane sulfonic acid (HEPES), dimethyl sulfoxide (DMSO), Tween-20, NHS, EDC and PEG, starch, saporin, bovine serum albumin (BSA), ASF, and ConA were obtained from Sigma-Aldrich (St. Louis, MO, USA). Sodium chloride and sodium bicarbonate were obtained from Sinopharm Chemical Reagent Co., Ltd. (Shanghai, China). Infant formula, infant cereal, lactase-containing milk powder, baby talcum powder, and protein powder were commercially purchased. Stevioside was purchased from a local supermarket in The Hague, Netherlands. Peach juice drink (juice content greater than or equal to 10%) was purchased from a local supermarket in Beijing, China. Ultrapure water (18 MΩ·cm) was generated by a Milli-Q A10 water purification system (Millipore, Bedford, MA, USA), and all other chemical reagents were of analytical grade.

Sample pads, absorbent pads, conjugate pads, and polyvinyl chloride (PVC) backing plates were purchased from Shanghai Jieyi Biotechnology Co., Ltd. (Shanghai, China). Nitrocellulose (NC) membrane was purchased from Sartorius (UniSart CN140, Göttingen, Germany). All glassware was soaked overnight in aqua regia and then thoroughly cleaned with ultrapure water three times, then dried for subsequent use. SERS spectra were measured using the portable Raman spectrometer (BWS 415-785H, B&W Tek, Newark, DE, USA). Samples for TEM imaging were sent to the Shanghai Moyan Testing Technology Centre. Zeta potential was tested using the nano-size measuring instrument NANO-ZS90 (Malvern, UK). The UV absorption wavelengths of SERS tags were detected using a UV-Vis spectrophotometer (Cray 300, Agilent Technologies Inc., Santa Clara, CA, USA), and the test strips were prepared using a film-stripping and gold-sputtering device (Biodot xyz5050, Shanghai JieNing Bio-Tech Co., Ltd. (Shanghai, China)), a paper cutter (Deli Group, Ningbo, China), and a strip cutter (ZQ2000, Shanghai Kinbio Tech. Co., Ltd., Shanghai, China).

**Caution:** Ricin, abrin, RCA120, AAG, Stx1, Stx2, SEB, RTA, and saporin are classified as highly toxic toxins. All associated experiments should be conducted following biosafety guidelines. Work should be performed within a fume hood during all procedures, and personal protective equipment, including gloves and safety goggles, should be worn. After the experiment, samples containing biotoxins should be thoroughly decontaminated by soaking in 5% NaOH overnight or autoclaving at 121 °C and 0.12 MPa for 20 min.

### 5.2. Synthesis of SERS NanoGPtags

AuNPs were prepared according to the Frens method [39] using sodium citrate reduction of chloroauric acid. In total, 100 mL of 0.01% (*w*/*v*) HAuCl_4_ was added to a clean three-necked round-bottomed flask, and a condensation tube was connected. A magnet was placed within the flask, and the rotation speed was set to 1100 rpm. The mixture was then heated and stirred in an oil bath at 115 °C, reaching boiling point for 30 min. Afterward, 5.0 mL of 1% (*w*/*v*) sodium citrate was introduced all at once, and the heating and stirring continued for an additional 40 min. The heating was stopped, and the rotation speed was adjusted to 600 rpm. Once the solution had cooled to room temperature, it was dispensed into 50 mL centrifuge tubes and stored at room temperature for use within a month.

The size of the AuNPs could be adjusted by changing the added volume of sodium citrate. For instance, by adding 0.94 mL of sodium citrate, AuNPs with a particle size of approximately 50 nm were prepared. Then, 10 mL of the freshly prepared 50 nm AuNPs was taken and added to a solution with a final concentration of 0.1 mM DTNB. This mixture was shaken at 800 rpm for 4 h. Subsequently, it was centrifuged at 6000 rpm for 10 min to remove the supernatant. The nanoparticles were washed twice with ultrapure water and then redissolved in 2 mL of 1.8 mM K_2_CO_3_ solution, forming AuNPs@DTNB.

We followed the method of Ma et al. [40] for covalent attachment of glycoprotein on the surface of AuNPs, with slight modifications. In total, 40 μg of ASF (1 g/L, dissolved in PBS) and 20 μL of PEG linker (0.5 g/L, dissolved in DMSO) were added to 440 μL of 50 mM NaHCO_3_ solution (pH 8.0). This mixture was incubated overnight at room temperature with gentle rotation, allowing for the successful conjugation of ASF to the PEG linker surface, forming the ASF-PEG7-S-S-PEG7-ASF complex. Subsequently, 1.5 mL of freshly prepared AuNPs@DTNB was introduced into the ASF-PEG7-S-S-PEG7-ASF solution. This was incubated at room temperature with rotation for 4 h, followed by freezing at −20 °C for 2 h [19]. Once the frozen sample was retrieved, it was redissolved at room temperature, and centrifugation was performed to remove the supernatant. The resulting pellet was washed twice with ultrapure water and redissolved in 100 μL of HBS-BT buffer solution (0.01 M HEPES, 0.15 M NaCl, 10 g/L BSA, 0.05% Tween-20, pH 7.4). The reconstituted complex was subsequently stored at 4 °C for future applications.

### 5.3. Preparation of Toxin-Specific SERS–LFA Strips

SERS-LFA strips prepared by the wet method comprised an absorbent pad, an NC membrane with a pore size of 8 μm, a conjugate pad, and a sample application pad. ConA and ASF antibodies were diluted in 0.01 M PBS and subsequently sprayed onto the control and test line of the NC membrane using a film-stripping and gold-sputtering device. The concentrations were 0.5 g/L for ConA and 1 g/L for ASF antibody. The strips were then air-dried at 37 °C for 2 h. Following this, the absorbent pad, conjugate pad, and sample application pad were assembled sequentially on a PVC backing sheet with the functionalized NC membrane. The assembly was then cut into multiple strips, each with a width of 3 mm. The test strips were ultimately stored in self-sealing bags with desiccant for future use. Before SERS-LFA detection, the SERS nanoGPtags had to be mixed with the sample.

In contrast, SERS-LFA strips prepared by the dry method included an absorbent pad, an NC membrane with a pore size of 8 μm, a conjugate pad with SERS nanoGPtags, and a sample application pad. The SERS nanoGPtags were firstly applied onto a binding pad with a 15 cm length, after which the preparation underwent vacuum freeze-drying for 6 h. Before use, the binding pad was cut into conjugate pad pieces of 1 cm in length. The dry method preparation was similar to the wet test strip preparation, except for the SERS-LFA detection process. In this case, the sample was directly applied to the test strip with the SERS nanoGPtags attached.

### 5.4. Optimization of Experimental Parameters

In the composition of SERS nanoGPtags, the concentration of ASF was fixed at 40 μg based on previous research, and the concentration of the ASF antibody in the C line was set at 1 g/L [13]. The main optimizations included the concentration of the T line coating, the amount of SERS nanoGPtags added to the wet test strip, the power and integration time for signal acquisition, and the amount of SERS nanoGPtags added to the dry test strip.

To optimize the T-line coating concentration, the concentration of ConA was set at 0.5, 0.7, 1, and 2 g/L and then applied to the NC membrane test area using a sprayer. Then, 1 g/L ASF antibody was applied to the control area. After assembling the test strip, it was immersed in 80 μL of 0.5% PBST (0.01 M PBS, 0.5% Tween-20, pH 7.4) solution to observe whether there was a false-positive band on the T line, thereby determining the optimized concentration.

The amount of SERS nanoGPtags added to the wet test strip was set at 1, 2, 4, 6, and 8 μL, mixed evenly with a final concentration of 100 ng/mL abrin solution, and the test strip was inserted. The SERS response of the T line was measured; 0.5% PBST mixed with the corresponding gradient concentration of the SERS nanoGPtags served as a blank control to observe whether there was a false-positive band, thereby determining the optimized amount to be added.

For the power and integration time of signal acquisition, with a fixed integration time of 15 s and an abrin concentration of 100 ng/mL, the acquisition power was set at 60, 90, 120, 150, and 180 mW. Also, with a fixed power of 90 mW and an abrin concentration of 100 ng/mL, the integration time was separately set at 5, 10, 15, 20, 25, 30, and 40 s. The optimized acquisition time and the best laser power were determined by comparing the SERS response of the T line.

The amount of SERS nanoGPtags added to the dry test strip was set at 1, 2, 2.5, 3, 4, and 5 μL, followed by freeze-drying and subsequent immersion in 80 μL of a buffer solution with 100 ng/mL abrin for SERS response measurement of T line. A blank control was used to detect potential false positives.

### 5.5. SERS-LFA Test Strip Assay

To improve the sensitivity and reliability of the test strips, the following parameters were optimized: the coating concentration of the T line, addition volume of SERS nanoGPtags, laser power, and acquisition time. Concentrations of ConA in the T line on NC membranes (0.5–2 g/L) and ASF antibody (1 g/L) on the control line were evaluated for potential false positives using test strips soaked in 0.5% PBST. The optimized volume of SERS nanoGPtags was determined by testing various volumes (1–8 μL) in a 100 ng/mL abrin solution against a blank control. Acquisition parameters were assessed through a range of laser powers from 60 to 180 mW and integration times from 5 to 40 s. Dry test strips to which SERS nanoGPtags (1–5 μL) had been applied were freeze-dried and immersed in a 100 ng/mL abrin buffer solution, with blank controls being used to verify the absence of false positives. For detection, samples were diluted in 2 mL EP tubes (to a final volume of 80 μL) using 0.5% PBST. A test strip was inserted into the EP tube to allow all liquid to be absorbed and migrated along the strip. After 15 min, the strip was dried in a 37 °C oven for 10 min to detect the SERS signal of the T line using a portable Raman spectrometer.

### 5.6. Evaluation of SERS-LFA Test Strips

Standard curves were established to correlate the SERS intensity of the T-line response with the concentrations of abrin and ricin. To evaluate the interference resistance of the test strip in the presence of white powder, abrin at a concentration of 100 ng/mL was introduced into a series of white powder solutions. Each solution was prepared in triplicate. The specificity of the test strip for abrin was assessed against RCA120, AAG, RTB, RTA, Stx1, Stx2, saporin, and SEB. The intra-batch and inter-batch reproducibility were determined using test strips with samples spiked with abrin and ricin. This method was then applied for the detection of authentic samples.

### 5.7. Real Sample Preparation

Crude abrin was obtained from several seed kernels by a process involving grinding, 5% acetic acid extraction, and subsequence galactose affinity purification [41]. The extracts from the peel of unripe castor fruits and the extracts from castor flowers were obtained by centrifugation of 2 g of ground powder in 2 mL of 0.01 M PBS extraction solution.

## Figures and Tables

**Figure 1 toxins-16-00312-f001:**
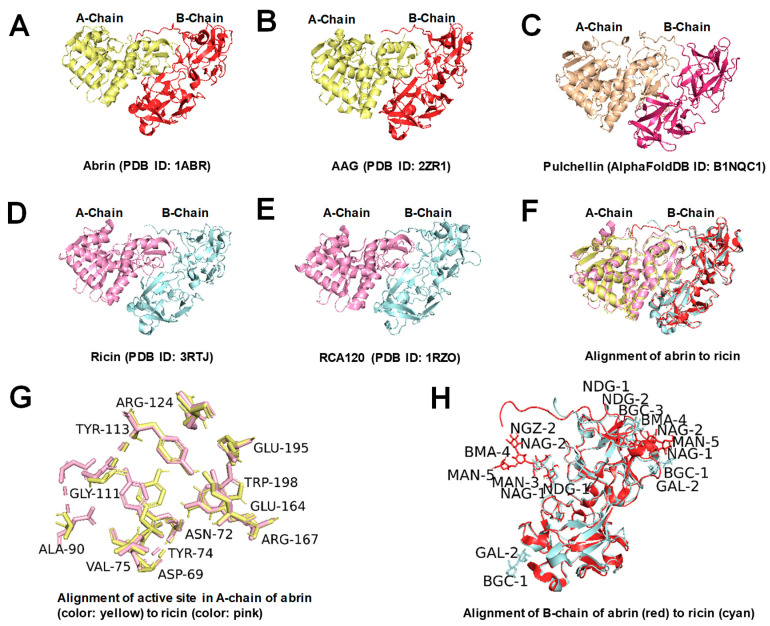
Three-dimensional structures of abrin, ricin, and similar proteins (drawn using PyMOL 2.5.4). (**A**–**E**) Three-dimensional structures of abrin, AAG, pulchellin, ricin, and RCA120, respectively; (**F**) alignment of abrin to ricin; (**G**) alignment of the active site in A-chain of abrin [3] (color, yellow) to ricin (color, pink); and (**H**) alignment of B-chain of abrin (color, red, with glycans including one NDG-NDG-BGC-BMA-MAN glycan chain and one NDG-NGZ-MAN-BMA-MAN glycan chain) to ricin (color, cyan, with glycans including two NAG-NAG glycan chains and 2 BGC-GAL glycan chains).

**Figure 2 toxins-16-00312-f002:**
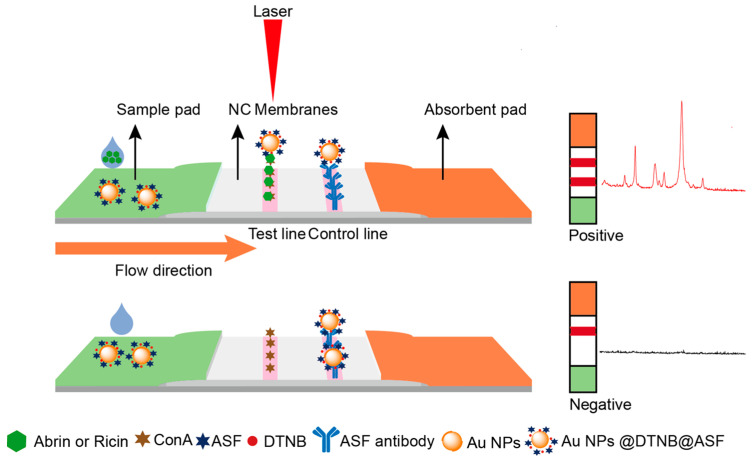
SERS-LFA diagram showing abrin and ricin detection via the glycoprotein sandwiched format.

**Figure 3 toxins-16-00312-f003:**
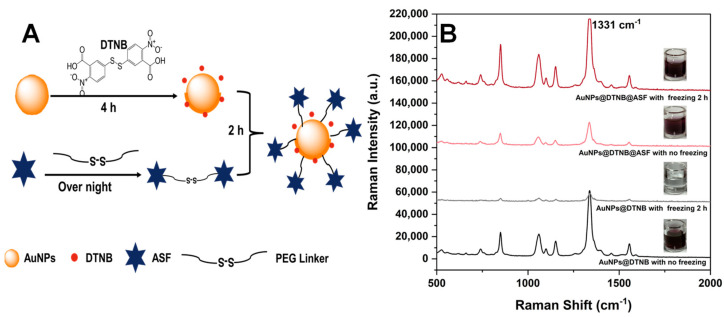
The preparation of SERS nanoGPtags. (**A**) SERS nanoGPtag fabrication process flowchart; (**B**) SERS response of AuNPs@DNTB and AuNPs@DNTB@ASF before and after freezing, with corresponding images inset.

**Figure 4 toxins-16-00312-f004:**
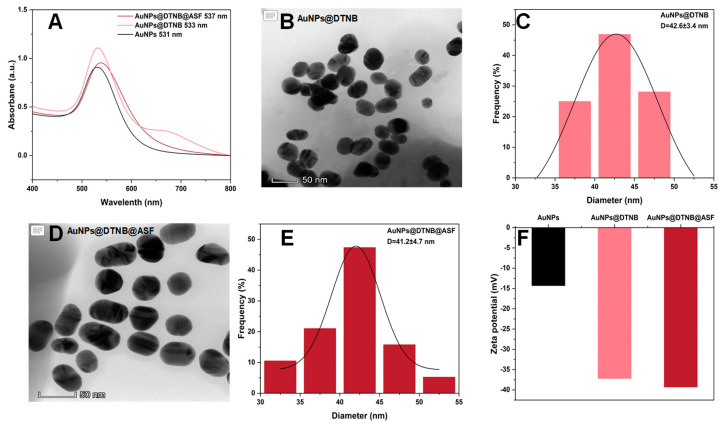
The characterization results of prepared SERS nanoGPtags. (**A**) The UV-Vis absorption spectra; (**B**,**C**) the TEM images and particle size distribution of AuNPs@DTNB; (**D**,**E**) the TEM images and particle size distribution of AuNPs@DTNB@ASF; (**F**) the Zeta electric potential results.

**Figure 5 toxins-16-00312-f005:**
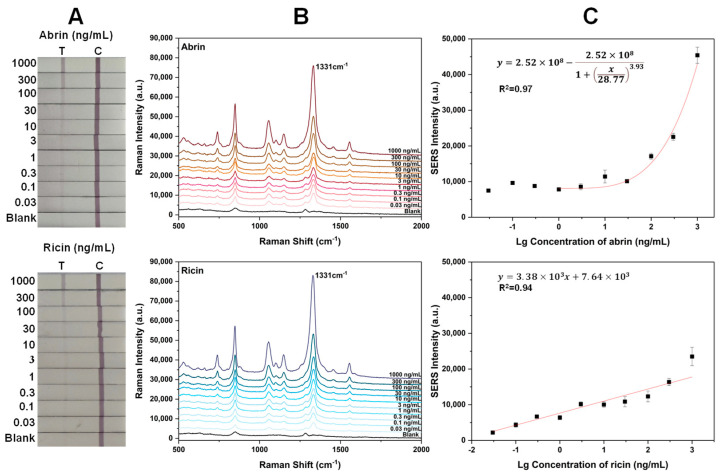
(**A**) Photographs of SERS-LFA strips for different concentrations of abrin and ricin; (**B**) SERS spectra measured upon the corresponding T lines; (**C**) plot of the SERS signal intensity at 1331 cm^−1^ as a function of toxin concentration via the wet method. The error bars represent the standard deviations of six measurements on three T lines.

**Figure 6 toxins-16-00312-f006:**
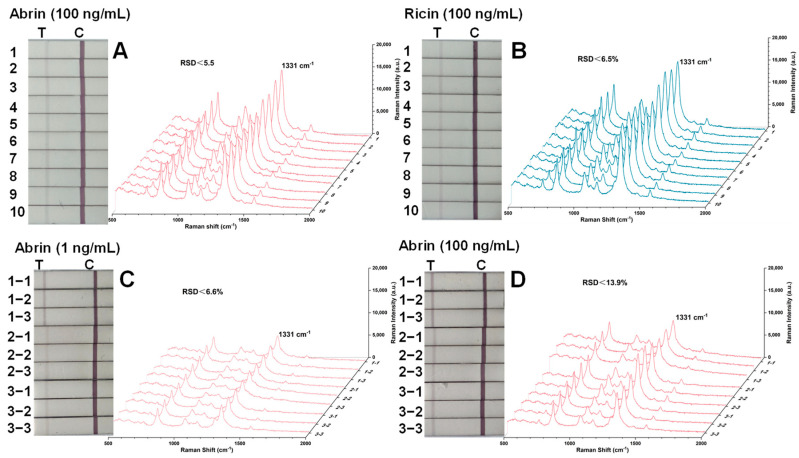
SERS spectra were measured on the T lines of specific SERS-LFA strips for repeatability measurements. (**A**,**B**) Repeatability of the test strip in batch for testing abrin and ricin; (**C**,**D**) repeatability between three batches of strips for testing abrin.

**Figure 7 toxins-16-00312-f007:**
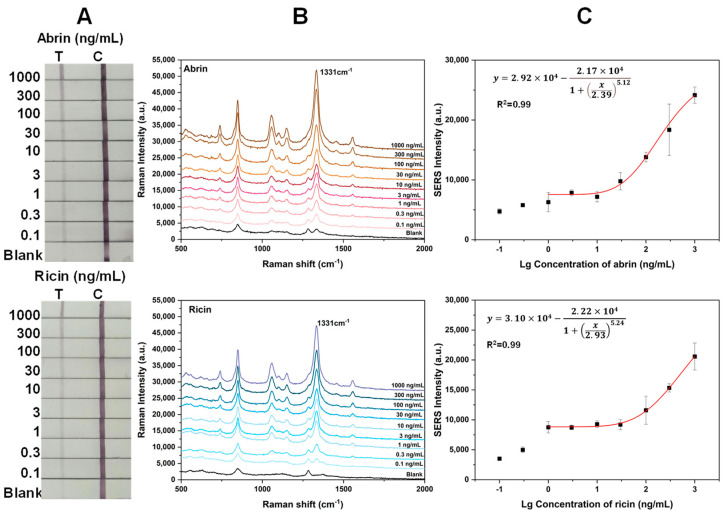
(**A**) Photographs of SERS–LFA strips for detecting different concentrations of abrin and ricin; (**B**) SERS spectra measured in the corresponding T lines; (**C**) plot of the SERS signal intensity at 1331 cm^−1^ as a function of toxin concentration via the dry method. The error bars represent the standard deviations of six measurements on three T lines.

**Figure 8 toxins-16-00312-f008:**
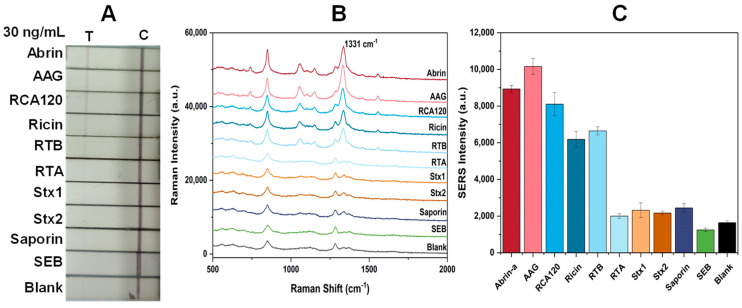
Specificity test with SERS–LFA strips. (**A**) Photographs of SERS–LFA strips for detecting abrin, AAG, RCA120, ricin, RTB, RTA, Stx1, Stx2, saporin, SEB, and blank control; (**B**) SERS spectra measured on the T lines of specific SERS-LFA strips for examining different proteins and a blank control; (**C**) histograms of SERS signal strength values at 1331 cm^−1^. The error bars are the standard deviations of six measurements on three T lines.

**Figure 9 toxins-16-00312-f009:**
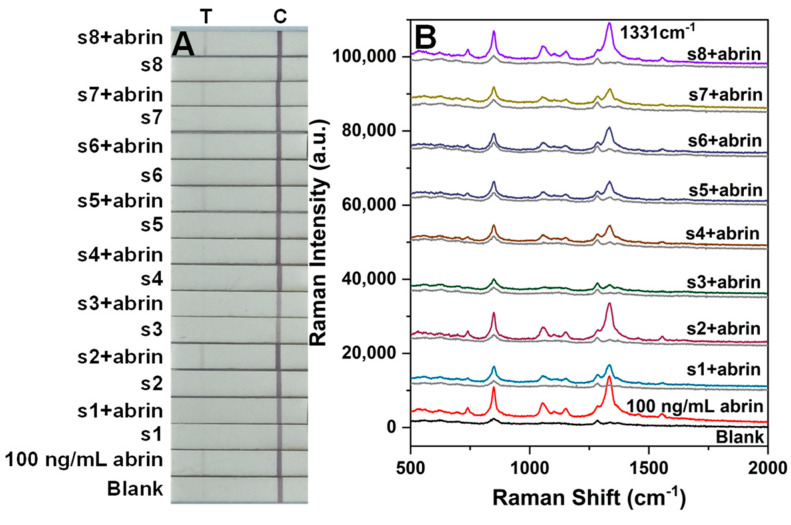
SERS-LFA assay for abrin spiked in different white powder solutions. (**A**) Photographs of SERS–LFA strips for detecting abrin in different white powders and blank control; (**B**) SERS spectra measured on the T lines of specific SERS-LFA strips for examining abrin in different white powders and a blank control.

**Figure 10 toxins-16-00312-f010:**
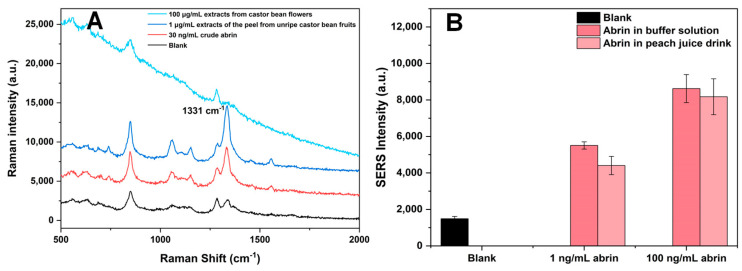
SERS-LFA assay spiked in actual samples. (**A**) SERS spectra measured on the T lines of specific SERS-LFA strips for examining actual botanic samples and a blank control; (**B**) histograms of SERS signal strength for examining values abrin-spiked peach juice drink sample at 1331 cm^−1^. The error bars are the standard deviations of six measurements on three T lines.

**Table 1 toxins-16-00312-t001:** SERS-LFA assay spiked in different white powder solutions.

White Powder	Negative	Positive
0.5%PBST (Control)	−	+
s1: Baby Milk	−	+
s2: Baby Talcum Powder	−	+
s3: Albumin Powder	−	+ (5 μg/mL)
s4: Milk Powder with Lactase	−	+
s5: Baby Rice Paste	−	+
s6: Stevioside	−	+
s7: Polyethylene Glycol (PEG)	−	+
s8: Starch	−	+

**Table 2 toxins-16-00312-t002:** An overview of recently reported methods for determining abrin or ricin.

Materials Used	Biocomponents	Target Protein	LOD	Reference
microfluidic chip	antibody	abrin, ricin	0.01 ng/mL (buffer)	[27]
microfluidic chip with a gold film	antibody	abrin	0.1 ng/mL (buffer)	[28]
AgNPs	aptamer	ricin B chain	10.2 fg/mL (buffer)	[29]
AgFON	aptamer	ricin B chain	4 μg/mL (blood)	[30]
silver dendrite nano substrates	antibody	ricin	4 μg/mL (milk)	[31]
AuNPs	antibody	ricin	1 ng/mL (buffer) 4 ng/mL (plasma)	[32]
AuNPs	oligodeoxynucleotides poly(21dA)	ricin	8.9 ng/mL (buffer)	[33]
silver dendrites	/	ricin B chain	4.4 μg/mL (water)	[34]
silver dendrites	aptamer	ricin B chain	10 ng/mL (buffer) 50 ng/mL (juice) 100 ng/mL (milk)	[35]
gold film over nanospheres	N-acetyl-galactosamine glycopolymer	ricin B chain	20 ng/mL (water)	[36]
SiO_2_@Au NPs (SERS-LFIA)	antibody	ricin	0.1 ng/mL (buffer)	[13]
AuNPs	glycoprotein	abrin, ricin	0.1 ng/mL (abrin in buffer) 0.3 ng/mL (ricin in buffer)	Our work

## Data Availability

Data are contained within the article or Appendix A.

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
