# Peer review of "A Glycoprotein-Based Surface-Enhanced Raman Spectroscopy–Lateral Flow Assay Method for Abrin and Ricin Detection"

_toxins, 2024, doi:10.3390/toxins16070312_

Round 1

Reviewer 1 Report

Comments and Suggestions for Authors

The manuscript submitted to toxins reports on the detection of abrin and ricin by a SERS-LFA method based on affinity recognition of the toxin by a pair of glycoproteins, i.e. asialofetuin and concanavalin.  The authors describe the preparation of the SERS tag with asialofetuin, the optimization of the experimental conditions for SERS-LFA prepared in dry and wet conditions, and the evaluation of the SERS-LFA on other proteins, white powders and botanic samples.

I found the manuscript interesting and well-written. However, I have major issues regarding the specificity of the LFA:

-          Asialofetuin and concanavalin bind to ricin and asialofetuin but can not distinguish between ricin and abrin. An additional detection methodology would be needed and added to the LFA system for selective ricin or abrin detection. In the present version of the test, this is not achieved.  

-          More importantly, the SARS-LFA would detect any ribosome-inactivating protein (RIP) with a lectin B chain. In my opinion, the current assay is not specific.  I noticed that the proteins tested in the Specificity section (2.7) are not RIP lectin and therefore not susceptible to bind the glycoproteins. I would test for instance viscumin and similar plant lectins.

-          The LFA detects the B chain of abrin and ricin, but only the entire toxin with A and B chains is active and toxic to humans. Therefore, the LFA detects inactive and active toxins.  Additional experiments would be needed in the situation of a positive result.

-          I was also surprised in Figure 7 by the low difference in signal between abrin and the undetected proteins. Abrin was spiked at 30 ng/mL which was 100 times the LOD. The detected signal must be 100 times higher for abrin when compared to undetected proteins, and not only 4 times higher (see Figure 7).

I would therefore not recommend the publication of the manuscript in Toxins.

Reviewer 2 Report

Comments and Suggestions for Authors

The presented ms “A glycoprotein-based SERS-LFA method for abrin and ricin detection” describes the binding of abrin and ricin to the glycoproteins asialofetuin (ASF) and concavalin A (ConA) with a detection limit of 0.1 for abrin and 0.3 ng/mL for ricin and their detection (abrin-ASF/ConA or ricin-ASF/ConA complexes through Surface-enhanced Raman Spectroscopy in combination with an Lateral Flow Assay (SERS-LFA).

They fabricated a sandwich format using asialofetuin (ASF) and concanavalin A (ConA) in our SERS-LFA test strip. These test strips have achieved rapid and highly sensitive detection of abrin and ricin and have been applied to the screening of  powder samples and liquid (juice drink) and  real plant samples.

Overall, the ms is sound and shows a straightforward and elegant application of their method for the detection of abrin and ricin contamination in food or other sample of interest. The reported detection limit is comparable to the limit recently reported by Bai et al.“A Self-Driven Microfluidic Chip for Ricin and Abrin Detection” (https://www.ncbi.nlm.nih.gov/pmc/articles/PMC9101213/).

Though this work is a proof of principle I miss a notion within the discussion on how to implement this detection method on a large scale for diagnostics and or commercialization of the test-strip and detection equipment. 

Reviewer 3 Report

Comments and Suggestions for Authors

The authors develop a SERS-LFA detection method using sandwich complex of glycoproteins instead of antibodies to detect ricin and abrin. Although the authors scientifically offer some useful insights, the English language used in this manuscript is inferior and fails to generate interest and/or maintain clarity for the readers. All throughout the manuscript, authors need to,

1. Use grammatically correct sentences
2. Avoid forming long sentences with incorrect punctuation
3. Avoid using additional/inappropriate filler words which do not enhance the message

Comments on the Quality of English Language

The English language used for writing this manuscript is scientifically below par and must be improved. All throughout the manuscript, authors need to,

1. Use grammatically correct sentences
2. Avoid forming long sentences with incorrect punctuation
3. Avoid using additional/inappropriate filler words which do not enhance the message

Reviewer 4 Report

Comments and Suggestions for Authors

The manuscript presents a glycoprotein-based SERS-LFA for the rapid and sensitive detection of abrin and ricin, which are type II ribosome-inactivating proteins (RIP II). The authors propose a method that uses glycoproteins as affinity recognition molecules, demonstrating good sensitivity, reproducibility, and rapid detection within 15 minutes.

Evaluation of the Manuscript:

  1. Discussion of Non-Plant Biotoxins:
    • Issue: The manuscript highlights the importance of detecting plant biotoxins (RIP type II) concerning biosecurity but does not discuss non-plant biotoxins. The authors should include a section describing non-plant biotoxins such as bacterial toxins (e.g., botulinum toxin) and explain their relevance to biosecurity. This addition will provide a broader context for the importance of biotoxin detection.
  2. Description of AB Toxin Structures:
    • Issue: The manuscript provides limited details on the structures of abrin and ricin and does not include a thorough comparison of their glycoprotein structures. A detailed structural comparison is missing, which is crucial for understanding the importance of glycoprotein-based LFAs. The authors should add a detailed comparison of abrin and ricin glycoproteins in the introduction and discussion sections. This comparison should cover the A domain’s N-glycosidase activity and the B domain’s lectin properties, leading to a better understanding of the glycoprotein-based LFA.
  3. Other RIP Type II Plant Biotoxins:
    • Issue: Other RIP type II plant biotoxins with similar structures to abrin and ricin, such as agglutinin and pulchellin, are not discussed. The manuscript does not mention these toxins, which could have therapeutic applications beyond biosecurity concerns. The authors should include a discussion of other RIP type II plant biotoxins like agglutinin and pulchellin in the introduction. These two plant toxins from the list could be described in introduction; such as Agglutinin and Pulchellin. The immunotoxicity of ricin and pulchellin are compared as anti-HIV immunotoxins, that could be described in the introduction. In addition, the similarity in the B-chain lectin of these plant toxin may results in the cross reactivity.
  4. Specificity Concerns (See Fig-7 and section 2.7):
    • Critics: The use of ConA as a lectin for capturing plant toxins raises concerns about potential cross-reactivity with other plant AB toxins such as Agglutinin and Pulchellin. Additionally, there is no discussion on why Saporin (a single-chain toxin) is not detected. The authors should address potential cross-reactivity with other plant AB toxins and clarify why their proposed LFA does not detect Saporin. This explanation should address possible cross-interactions and specificity issues.
  5. Sensitivity Discussion:
    • Critics: The limit of detection (LoD) for ricin using the proposed SERS-LFA method is 0.3 ng/mL. The adequacy of this sensitivity for biosecurity purposes is not discussed. The manuscript should discuss whether this sensitivity is sufficient for biosecurity concerns. The authors should provide a detailed discussion on the adequacy of the LoD for ricin (0.3 ng/mL) in the context of biosecurity. This discussion should compare it with other detection methods and relevant biosecurity thresholds.
  6. Applicability to Degraded Toxins:
    • Issue: The manuscript does not address whether the method can detect degraded or structurally altered biotoxins, which is important for real-world biosecurity applications. The authors should include a discussion on the method’s applicability to detect degraded or structurally compromised toxins. Emphasizing the detection of inactivated or degraded biotoxins will highlight the method’s relevance for biosecurity scenarios.

By addressing these recommendations, the manuscript will provide a more comprehensive and informative discussion on the detection of plant toxins and their implications for biosecurity.

Comments on the Quality of English Language

Minor editing of English language required.

Round 2

Reviewer 1 Report

Comments and Suggestions for Authors

Comment 1: ok

Comment 2: In this context, the authors should state that the assay might give a positive response for viscumin and any RIPII toxin able to bind the glycoproteins

Comment 3: In my opinion, the new results do not indicate a differenciation of inactive and active ricin because a signal is detected for both and a difference of 1.5 is minor. 

Comment 4: I checked figures 7 and 8. The authors have to  explain why the signal increased between 0.1 and 1 ng/mL, then remained constant between 1 and 50 ng/mL, and then increased again between 50 and 1000 ng/mL

Author Response

Response to the comments from Referee 1:

Comment 1: ok

Response: Thank you, we appreciate it.

Comment 2: In this context, the authors should state that the assay might give a positive response for viscumin and any RIPII toxin able to bind the glycoproteins

Response: Thanks for your constructive suggestion. We added this point to the revised version; please refer to lines 356-361, page 11.

Comment 3: In my opinion, the new results do not indicate a differenciation of inactive and active ricin because a signal is detected for both and a difference of 1.5 is minor.

Response: Thank you for your suggestion. Our method can distinguish between active and inactive toxins to some extent, but a more accurate distinction is not yet possible, which can be followed up by combining the N-glycosidase activity of the A-chain to achieve the distinction between active and inactive toxins. We have already addressed this aspect in the corresponding section; please refer to lines 251-254, page 8.

Comment 4: I checked figures 7 and 8. The authors have to explain why the signal increased between 0.1 and 1 ng/mL, then remained constant between 1 and 50 ng/mL, and then increased again between 50 and 1000 ng/mL

Response: Our curve is fitted using the logistic function, which slows down its growth as it approaches its asymptotic values (0 or 1). In the low concentration range (1 ng/mL to 30 ng/mL), the logistic function may not have reached its rapid growth phase yet, so the growth appears relatively slow. Additionally, there is a certain degree of experimental error; in the low concentration range, measurement errors may have a greater impact on the results, making the curve appear to grow slowly. In contrast, in the high concentration range, the influence of measurement errors may be relatively smaller due to the increase in signal strength. Therefore, we have demonstrated that the calculated inactive protein content is not so appropriate. Please refer to lines 252-256, page 8. 

Reviewer 4 Report

Comments and Suggestions for Authors

The authors have addressed all the concerns regarding the specificity and sensitivity of their proposed SERS-LFA biosensor.

This new version is appropriate for publication.

Comments on the Quality of English Language

Minor editing of English language required

Author Response

Response to the comments from Referee 4:

Comments and Suggestions for Authors

The authors have addressed all the concerns regarding the specificity and sensitivity of their proposed SERS-LFA biosensor.

This new version is appropriate for publication.

Comments on the Quality of English Language

Minor editing of English language required

Response: Thanks for your kind support of our manuscript. We are very appreciative. The English language has been revised and polished one more time; please check the whole context.

Round 3

Reviewer 1 Report

Comments and Suggestions for Authors

Additional point: in the revised version of the conclusion section, the sentence "we suggest that this detection method can be implemented on a large scale for diagnostics..." was added. Serum or plasma was not evaluated in this manuscript, so application to diagnostic can not be claimed by the authors. Please remove.     

Author Response

Additional point: in the revised version of the conclusion section, the sentence "we suggest that this detection method can be implemented on a large scale for diagnostics..." was added. Serum or plasma was not evaluated in this manuscript, so application to diagnostic can not be claimed by the authors. Please remove.    

Response: Thanks for your constructive suggestion, we've removed this section and modified it to “this detection method can be implemented on a large scale for primary screening.” please refer to lines 401, page 12.